# Rapid in situ carbon-13 hyperpolarization and imaging of acetate and pyruvate esters without external polarizer

Obaid Mohiuddin[1], Henri de Maissin [1,2], Andrey N. Pravdivtsev [3], Arne Brahms [4], Marvin Herzog[1,2], Leif Schröder [5], Eduard Y. Chekmenev [6], Rainer Herges[4], Jan-Bernd Hövener[3], Maxim Zaitsev[1], Dominik von Elverfeldt [1] & Andreas B. Schmidt [1,2,6] ✉

Hyperpolarized $^{13}$C MRI visualizes real-time metabolic processes in vivo. In this study, we achieved high $^{13}$C polarization in situ in the bore of an MRI system for precursor molecules of most widely employed hyperpolarized agents: [1-$^{13}$C]acetate and [1-$^{13}$C]pyruvate ethyl esters in their perdeuterated forms, enhancing hyperpolarization lifetimes, hyperpolarized to $P_{13C} \approx 28\%$ at 80 mM concentration and $P_{13C} \approx 19\%$ at 10 mM concentration, respectively. Using vinyl esters as unsaturated Parahydrogen-Induced Polarization via Side-Arm Hydrogenation (PHIP-SAH) precursors and our novel polarization setup, we achieved these hyperpolarization levels by fast side-arm hydrogenation in acetone-$d_6$ at elevated temperatures (up to 90°C) and hydrogenation pressures (up to 32 bar). We optimized the hyperpolarization process, reducing it to under 10 s, and employed advanced pulse sequences to enhance the polarization transfer efficiency. The hyperpolarization system has a small footprint, allowing it to be positioned in the same magnet, where $^{13}$C MRI is performed. We exemplified the utility of the design with sub-second in situ $^{13}$C MRI of ethyl [1-$^{13}$C] pyruvate-$d_6$. However, challenges remain in side-arm cleavage and purification in the MRI system to extract highly polarized aqueous agent solutions. Our results showcase efficient and rapid $^{13}$C hyperpolarization of these metabolite precursors in an MRI system with minimal additional hardware, promising to enhance future throughput and access to hyperpolarized $^{13}$C MRI.

Non-invasive imaging of the human anatomy is an important pillar of diagnostic medicine. Morphological criteria are heavily used to diagnose and grade disease, as exemplified by the RECIST criteria for tumor assessment[1]. However, they hold limitations in capturing early molecular responses, while changes may not emerge early enough in traditional size-based assessments, even when patients respond favorably[2]. Hence, the evolving field of oncology calls for non-radioactive methods that provide swift and rich insights into metabolism for cancer staging and grading, treatment selection and response analysis, and survival[3,4].

Magnetic Resonance Imaging (MRI) is a powerful and versatile modality with great promise for molecular imaging. Notably, Nuclear Magnetic Resonance (NMR) is a gold standard in analytical chemistry for determining structure and for discriminating between molecules enabled by chemically shifted resonance frequencies and line splitting due to spin-spin interactions between neighboring spins. However, the intrinsically low sensitivity of magnetic resonance has been a persistent challenge, limiting MRI mostly to detect $^1$H nuclei in water and lipids that are highly abundant in the mammalian body. In the quest to amplify the diagnostic potential of MRI, hyperpolarization techniques have garnered significant attention

[1]Division of Medical Physics, Department of Diagnostic and Interventional Radiology, University Medical Center Freiburg, Faculty of Medicine, University of Freiburg, Killianstr. 5a, 79106 Freiburg, Germany. [2]German Cancer Consortium (DKTK), partner site Freiburg, German Cancer Research Center (DKFZ), Im Neuenheimer Feld 280, Heidelberg, 69120, Germany. [3]Section Biomedical Imaging, Molecular Imaging North Competence Center (MOINCC), Department of Radiology and Neuroradiology, University Medical Center Schleswig-Holstein and Kiel University, Am Botanischen Garten 14, 24118 Kiel, Germany. [4]Otto Diels Institute for Organic Chemistry, Kiel University, Otto-Hahn-Platz 5, 24118 Kiel, Germany. [5]Division of Translational Molecular Imaging, German Cancer Research Center (DKFZ), Im Neuenheimer Feld 280, 69120 Heidelberg, Germany. [6]Integrative Biosciences (Ibio), Department of Chemistry, Karmanos Cancer Institute (KCI), Wayne State University, 5101 Cass Ave, Detroit, MI, 48202, USA. ✉e-mail: andreas.schmidt@uniklinik-freiburg.de

enabling MRI of selected molecules with over 10,000-fold signal enhancement[5,6]. The ability to hyperpolarize key metabolites such as pyruvate has propelled this interest by offering real-time insights into metabolic processes[3,4]. For such applications, non-$^1$H nuclei such as carbon-13 ($^{13}$C) are typically hyperpolarized (HP), because of their background-free detection in vivo, large chemical shift dispersion, and long lifetimes of HP states in vivo compared to proton (i.e. longitudinal relaxation time $T_1$ of 1 min for [1-$^{13}$C]pyruvate versus $\approx 1$ s for most protons). This endeavor has been supported by the identification of specific metabolic alterations delivering powerful surrogate markers for disease staging and evaluating progression and therapy response. HP MRI, particularly using HP [1-$^{13}$C]pyruvate, is currently under investigation in over 50 clinical trials, and interim results have shown significant promise for future patient care[5,7–9].

While pyruvate remains the most widely applied hyperpolarization agent, acetate probes aberrant fatty acid metabolism[10–12]. For instance, non-HP [1-$^{13}$C]acetate has been used in brain metabolism studies[13–17], and HP [1-$^{13}$C]acetate has been used in studies of liver and brain metabolism[13,18,19]. Hence, the efficient hyperpolarization of acetate is as important in the context of biomedical applications.

Biomedical applications, including all clinical studies, have relied on dissolution Dynamic Nuclear Polarization (d-DNP), which is recognized as the most established method for producing a variety of $^{13}$C HP metabolites[3,4,20]. The d-DNP hyperpolarization requires strong magnetic fields (>1 T, typically employing superconducting magnets), cryogenic temperatures (<2 K), and microwave irradiation to polarize agents in the solid phase. This is followed by rapid thawing and sample dissolution, typically using superheated water, to transfer HP metabolites into the liquid aqueous phase. This process often takes about one hour e.g. for producing a bolus of HP [1-$^{13}$C]pyruvate. The impressive results achieved with HP MRI have sparked a persistent effort within the scientific community to explore other technologies and to enhance d-DNP for a faster, more flexible, and more cost-effective production of HP media[21–26].

Parahydrogen-based techniques hyperpolarize in the liquid state using the readily available spin order of the singlet isomer of molecular hydrogen (parahydrogen, pH$_2$), often employing inexpensive hardware and non-elaborate sample preparation methods[24,25]. Hence, hydrogenative pH$_2$-Induced Polarization (PHIP)[27,28] and Signal Amplification by Reversible Exchange (SABRE, also known as non-hydrogenative PHIP)[25,29] have emerged as promising high-throughput alternatives to d-DNP both capable of producing agents swiftly within few minutes as recently exemplified with safe in vivo demonstrations[30–35].

Hydrogenative PHIP experiments can be classified into two fundamentally distinct subtypes: (a) Parahydrogen And Synthesis Allows a Dramatically Enhanced Nuclear Alignment (PASADENA), where hydrogenation, optional heteronuclear polarization transfer using pulse sequences, and readout occur at the high magnetic field[28,36,37]; b) Adiabatic Longitudinal Transport After Dissociation Engenders Net Alignment (ALTADENA), a magnetic field cycling (MFC) scheme, where hydrogenation takes place at low field followed by adiabatic transfer to higher magnetic fields for detection of the in-phase $^1$H spin order[38]. Similarly, an MFC can be used for heteronuclear polarization transfer by passaging ultra-low fields[39,40]. Until recently, hydrogenative PHIP for biomedical applications was severely limited in its choice of agents, primarily due to the scarcity of suitable unsaturated precursors that could undergo hydrogenation to produce metabolically active molecules[24,41]. However, PHIP by Side Arm Hydrogenation (PHIP-SAH) has made strides in enabling the hyperpolarization of various molecules including pyruvate and acetate that have no direct unsaturated precursor that can be hydrogenated with parahydrogen (pH$_2$)[32,40]. In PHIP-SAH experiments, in a three-step procedure, unsaturated esters of a metabolite of interest are hydrogenated typically in acetone or chloroform yielding PHIP-SAH precursor molecules of the desired metabolite, followed by side-arm cleavage, and a swift purification to extract the metabolite in water[32,42,43].

A variety of advanced instruments for PHIP has been developed, each offering distinct advantages, as detailed in a recent review[24]. Standalone polarizers were presented, many of the first used polysulfone reactors for hydrogenation in water, allowing for elevated pH$_2$ pressures and temperatures to enhance the hydrogenation rate[44–47]. However, water's limited solubility for H$_2$ and some PHIP-SAH esters, compared to organic solvents, restricts the concentration of agents. Additionally, polysulfone is incompatible with many organic solvents, such as acetone and chloroform. To overcome these limitations, 5-mm NMR tubes have been employed[48–52], which can tolerate some overpressure, or steel, brass and aluminum reactors have been used[53,54]. However, for hyperpolarization at high magnetic fields, metallic reactors are unsuitable because they shield radio frequencies and have much different magnetic susceptibilities compared to liquids, causing significant magnetic field inhomogeneity. Similar to d-DNP, all these hydrogenative PHIP implementations still require an additional device and investment for a "polarizer" to prepare the hyperpolarized (HP) agents for in vivo $^{13}$C MRI. The polarizer itself can be complex and require considerable space within the imaging facility in close proximity to the MRI system due to signal decay during sample transfer.

Our team has pushed forward a PASADENA variant that enables the ultra-fast polarization of $^{13}$C agents in situ within an MRI system with a minimized footprint: Synthesis Amid the Magnet Bore Allows Dramatically Enhanced Nuclear Alignment (SAMBADENA)[55]. The primary advantage of this method is the elimination of the need for an external polarizer, as the highly advanced MRI hardware, augmented with a "PHIP upgrade," to polarize agents directly in the application field. Consequently, SAMBADENA circumvents relaxation losses and the need for transporting agents to the MRI setup. However, the elevated magnetic field restricts the materials that can be used and, unlike external polarizers, the limited space available in preclinical MRI systems poses an additional engineering challenge for successful implementation.

In our previous work, we demonstrated the administration and in vivo imaging within seconds using [1-$^{13}$C]hydroxyethyl propionate-$d_3$ (HEP), a xenobiotic molecule hyperpolarized in water and administered without purification[56]. We showed that both the reactor and the mouse can be positioned at the isocenter of the MRI system for hyperpolarization and in vivo imaging, respectively, facilitated by a motorized slider moving the mouse bed and reactor in between the two experimental steps during the tail vein administration. To date, high $^{13}$C polarizations of several PHIP agents in water were demonstrated with SAMBADENA using custom polysulfone reactors, namely HEP, [1-$^{13}$C]succinate-$d_2$, and [1-$^{13}$C]phospholactate-$d_2$[55,57,58]. Additionally, high $^{13}$C-polarization throughput (hyperpolarization every 15 s) has been shown[59]. However, PHIP-SAH and the hyperpolarization of acetate and pyruvate esters have not been demonstrated with SAMBADENA yet due to challenges in synthetic chemistry, spin physics, and the development of a suitable reaction chamber.

Recent advances in the synthesis of suitable $^{13}$C and $^2$H isotope labeled PHIP-SAH precursors[60–64], and polarization transfer methods operating at clinical magnetic fields[65–68] have facilitated investigating this promising technique. PHIP-SAH at Tesla fields was predominantly pioneered by the Glöggler group, demonstrating impressively high $^{13}$C polarization levels of up to $P_{13C} \approx 50\%$ for ethyl [1-$^{13}$C]acetate-$d_6$ and for ethyl [1-$^{13}$C]pyruvate-$d_6$ inside a 300 MHz NMR system[43,69]. Recently, it has been suggested to use a low-cost benchtop NMR systems or a standalone polarizer operating in the millitesla regime to avoid necessitating an additional costly NMR system for polarization[65,70,71]. Still, these approaches require an external polarizer to produce HP metabolites.

In this study, we present a SAMBADENA approach that enabled $^{13}$C hyperpolarization up to $P_{13C} \approx 30\%$ for ethyl [1-$^{13}$C]acetate-$d_6$ and ethyl [1-$^{13}$C]pyruvate-$d_6$. The key innovation was the development of an efficient hydrogenation process at high field, in acetone-$d_6$, at elevated temperature $\geq 90°$C and pressure >30 bar that resulted in complete hydrogenation of unsaturated precursor under 10 s. This achievement was made possible by a newly designed organic solvent-compatible non-magnetic SAMBADENA setup, Fig. 1. While we omit the side-arm cleavage and purification steps typically used with PHIP-SAH for biomedical applications, we demonstrate the potential of our results by showcasing the rapid $^{13}$C MRI of 1 mL of

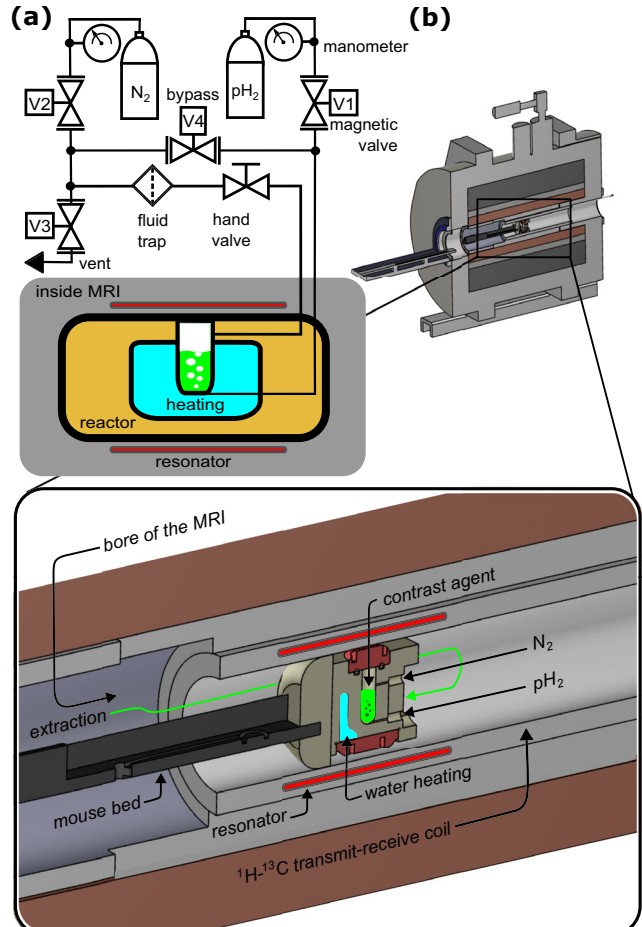

**Fig. 1 | Schematic of the SAMBADENA setup. a** A fluid control unit with two manometers, four electric solenoid valves (V1-V4), and one hand valve was used to inject the fluids and $pH_2$ at the bottom (labeled as V1 in Supplementary Fig. 1a) of the reactor and to create backpressure with $N_2$ or release pressure on top (labeled as V2 in Supplementary Fig. 1a). The liquid chamber was held in the isocenter of the MRI system. **b** Drawing of the SAMBADENA setup with magnification to show the new PEEK reactor installed in the MRI system (cross-sectioned). Note that this reactor is designed to be compatible with a mouse bed setup, as utilized in our previous in vivo experiments[56].

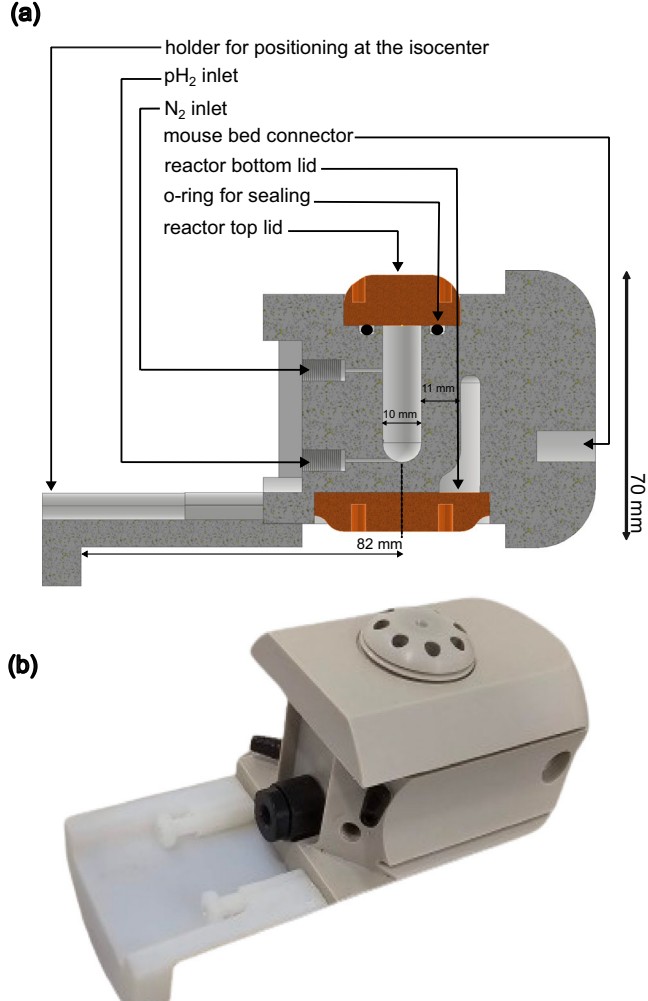

**Fig. 2 | Reactor used for SAMBADENA experiments. a** Cross-sectional view of the reactor designed using CAD. It illustrates the two-chamber structure, with the inner chamber designated for hosting the hydrogenation reaction and the outer chamber for active heating. **b** Photograph of the reactor, showing the top lid covering for the reaction chamber. Two inlets, one on each side (black), are connecting the heating chamber of the reactor for the hot water circulation. A one-way PEEK check valve (black) is connected to the bottom inlet of the reaction chamber to direct gas flow into the reactor. The white attachment connected to the reactor was used for positioning the reactor in the MRI system when used without the mouse bed.

80 mM of ethyl [1-$^{13}$C]pyruvate-$d_6$ in situ within seconds following the completion of the hyperpolarization process.

## Materials and methods

### Experimental setup

The SAMBADENA setup comprised a custom-made, solvent-compatible, actively-heated reactor for fast $pH_2$ addition, described in the next section. For hyperpolarization and MRI, a preclinical 7-T MRI setup (BioSpec 70/20, Advance III, Bruker, USA) and a $^1$H-$^{13}$C volume coil (V-XLS-HL-070, Rapid Biomedical, Germany) were used, Fig. 1.

The fluid control included a set of four magnetic valves (G052 with solvent-compatible perfluoroelastomer (FFKM) sealings, GSR Ventiltechnik, Germany), which were operated via a microcontroller-based relay box from the MRI pulse program that was described previously[59], and one hand valve. A non-magnetic, FFKM-sealed PEEK check valve (CV-3320, IDEX, USA) was installed in the reactor inlet at the bottom. A fluid trap was installed right before valve V3 to avoid contaminating the magnetic valves with solvents and catalyst. This fluid trap was additionally used as exhaust when tubes and the reactor were cleaned by flushing acetone at the end of each experiment day. This setup was automated and actuated via the MRI system's console.

### Reactor design

A reactor was custom-designed for this study (Inventor Professional, Autodesk, USA) and made from polyether ether ketone (PEEK) via computer numerical control (CNC), Fig. 2. PEEK was chosen because of good magnetic and mechanical properties making it robust against elevated pressures and temperatures, combined with excellent resistance to many acids, bases and solvents including water, acetone and chloroform. The reactor features two chambers: an inner chamber for the hydrogenation reaction and an outer chamber for active heating. The inner chamber has four inlets, two of which (one at the top, one at the bottom) are utilized in this study. The bottom inlet of the reactor facilitates $pH_2$ bubbling. The top inlet serves three main purposes: pre-pressurizing the reactor with nitrogen to prevent boil-off, depressurizing the reactor after each experiment, and for cleaning the setup by flushing acetone. The remaining inlets, intended for injection of the base for cleavage and extraction of the solution, were sealed for this study.

The reactor was engineered to withstand high pressures and a finite element method based numerical simulation suggested it can hold up to

162 bar with no permanent deformation (i.e. a pressure up to 50 bar can be applied with a safety factor larger three, Supplementary Fig. 2). Experimentally, we successfully tested the setup operation with up to $p = 32$ bar at $T = 90$ °C. Solvent compatible FFKM materials were used to seal the top and bottom lids of the reactor.

The reactor also included a heating chamber and was actively heated using a water-circulation system during the experiments (PC300 immersion circulator, ThermoFisher scientific, USA). At the beginning of an experiment day, the water heating was turned on for approximately 30 min before performing hyperpolarization experiments to ensure an equilibrium temperature for the reactor. After this initial heating, the reactor was used for polarization experiments at a rate of up to one sample per two minutes. The reactor was mountable in the MRI system either directly to the $^1$H-$^{13}$C coil using an attachment (see Fig. 2b), or to a mouse bed. For the experiments here, the reactor was attached to a mouse bed (Fig. 1b) and positioned in the

isocenter of the magnet using an automated positioning system (AutoPac, Bruker, USA).

Technical drawings of the reactor, cross sections, and a pressure simulation are provided in the SI.

### Sample preparation

Parahydrogen was enriched to ≈90%[72]. Reaction solutions were prepared by mixing vinyl acetate-$d_6$ (VA, ≈1.1% naturally abundant $^{13}$C, CAS: 189765-98-8, ID:UN1301, EQ Laboratories, Germany) or custom-synthesized vinyl [1-$^{13}$C]pyruvate-$d_6$ (VP, ≈98% $^{13}$C enriched[61] provider: Prof. Rainer Herges, QuantView GmbH, Germany) and a rhodium-based hydrogenation catalyst ([Rh(dppb)(COD)]BF4, CAS: 79255-71-3, ID:341134, Sigma Aldrich, USA) in 1 mL of acetone-$d_6$ (CAS: 666-52-4, ID:151793, Sigma Aldrich, USA). During the experiments, VA was hydrogenated to ethyl-acetate-$d_6$ (EA) and VP formed ethyl [1-$^{13}$C]pyruvate-$d_6$ (EP), Fig. 3a.

### Hyperpolarization process

To commence the experiment, 1 mL of the precursor catalyst solution was filled into the reactor (inside the MRI system), pressurized from the top of the reactor under a 5 bar $N_2$ atmosphere to avoid boil-off, and warmed using the reactor-integrated water heating for 10 s. Then, the hand valve at the reactor outlet was closed to limit the outlet volume filled during the pH$_2$ injection. The pH$_2$ addition was started by opening valve V1 for 7 s to inject pressurized pH$_2$ (≈90% enrichment) from the bottom of the reactor before the spin order transfer (SOT) sequence was started. The ESOTHERIC (efficient spin order transfer to heteronuclei via relayed INEPT (insensitive nuclei enhanced by magnetization transfer) chains) sequence was used with two composite refocusing pulses per time interval (90°$x$-180°$y$-90°x, Fig. 3b) for the SOT from pH$_2$-derived protons to $^{13}$C[71].

In total, the hyperpolarization process for a sample was completed in approximately 7.4 s, consisting of 7 s for hydrogenation followed by 388 ms for SOT. An additional 50 s time was allocated for sample filling and warming. Hence, with our technique, producing one hyperpolarized batch per minute becomes feasible.

### Signal detection and quantification

In most cases, the hyperpolarized $^{13}$C signal was detected at the end of the SOT by acquiring free induction decay of $^{13}$C signal. The polarization level $P_{13C}$ was quantified by comparing the Fourier-transformed $^{13}$C signals of the HP sample and an external reference sample at thermal equilibrium (4 M [1-$^{13}$C]sodium acetate in 1.5 mL H$_2$O, 99% $^{13}$C, CAS: 23424-28-4, ID: 279293, Sigma Aldrich, USA; doped with gadolinium (7.5 μL Vasovist), $T_1 = 832$ ms). In contrast to other studies[73], the direct detection of thermal-equilibrium $^{13}$C signal from our HP samples was not possible due to the relatively long $^{13}$C $T_1$ relaxation times and the low detection sensitivity of our volume resonator (see SI, section 8). We calculated $^{13}$C polarizations via the formula

$$P_{HP} = P_B \cdot \frac{N_{ref}}{N_{HP}} \frac{F_{13C-ref}}{F_{13C-HP}} \frac{c_{ref}}{c_{HP}} \frac{V_{ref}}{V_{HP}} \frac{S_{HP}}{S_{ref}} \qquad (1)$$

where "HP" and "ref" refers to the hyperpolarized and reference signal, respectively. $P$ is the polarization with $P_B \approx 6.0 \times 10^{-4}$% referring to the Boltzmann equilibrium polarization at 7 T at 300K, $N$ is the number of summated MR signal ($N_{HP} = 1$ in all cases), $F$ is the fraction of $^{13}$C isotope, $c$ is the concentration of the molecule, $V$ is the volume of the sample and $S$ is the measured signal. Here, $^{13}$C polarization level was quantified assuming 100% hydrogenation yield. The reaction kinetics investigated in Fig. 3d uses a two-handed fitting model to reveal time constants for the hydrogenation, $T_{hydr}$, and relaxation of the pH$_2$-derived $^1$H spin order before SOT, $T_{relax}$[55,68]:

$$P_{13C}(t_h) = P_{max} \cdot \left( (T_{hydr}/T_{relax}) - 1 \right)^{-1}$$
$$\cdot \left[ \exp\left( -(t_h - t_0)/T_{hydr} \right) - \exp\left( -(t_h - t_0)/T_{relax} \right) \right] \qquad (2)$$

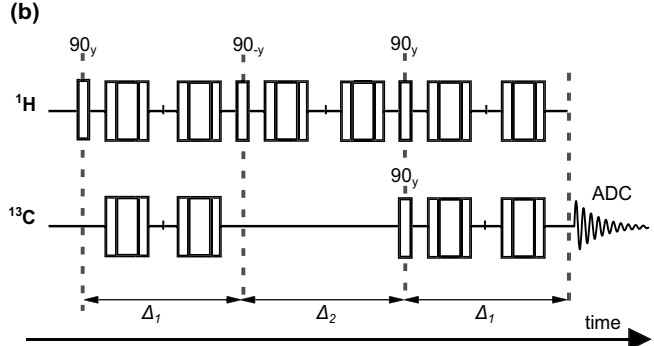

**(a)**

vinyl [1-$^{13}$C]acetate-$d_6$          vinyl [1-$^{13}$C]pyruvate-$d_6$

pH$_2$ + cat.

SOT

ethyl [1-$^{13}$C]acetate-$d_6$          ethyl [1-$^{13}$C]pyruvate-$d_6$

**(b)**

**Fig. 3 | Reaction schemes and SOT sequence diagram. a** Upon catalytic, pairwise addition of parahydrogen (pH$_2$) to the precursors (top) and SOT, the 1-$^{13}$C-hyperpolarized esters were obtained (bottom). **b** The ESOTHERIC sequence[66] was used with three 90° $^1$H- and one 90° $^{13}$C-pulses (single bars, y/-y phase) and two composite refocusing pulses in each free evolution interval (90°$x$-180°$y$-90°$x$) (Supplementary Fig. 5). The intervals $\Delta_1 = 1/(2J_{12})$ and $\Delta_2 = 1/(2J_{23})$ were calculated using the $J$-couplings of the three polarization-exchanging nuclei. The $J$-couplings were taken from the literature: for VA, $J_{12} = 6.94$ Hz, $J_{23} = 3.16$ Hz; and for VP, $J_{12} = 7.1$ Hz and $J_{23} = 3$ Hz[69]. $J_{13} = 0$ was assumed for both molecules.

$P_{max}$ is the maximum polarization value reached if no relaxation is present, $t_0$ is the time offset to consider time needed for the delivery and dissolution of $pH_2$, pressure build-up, and catalyst activation.

## $^{13}$C MRI

An MRI sequence and acquisition started subsequently after application of SOT with an additional 90°$_y$ $^{13}$C flip-back pulse (single-shot $^{13}$C RARE, 128x64x1 matrix size, 40×24 mm$^2$ field of view (FOV), 15-mm slice thickness, 0.31×0.38 mm$^2$ in-plane resolution, 64 echoes measured after one excitation, 438.6 ms acquisition time). For coregistration, an X-ray micro-tomography (μCT) image of the empty reactor was acquired to depict the setup (res.: 20 μm, exposure: 266 ms angular step: 0.8°, Bruker SkyScan 1276). Additional experimental details are presented in the supporting information (SI).

## Results and discussion

### Hyperpolarization of PHIP-SAH precursors

High 1-$^{13}$C polarizations of 28% and 19% (Fig. 4a) were obtained for 1 mL EA (80 mM, 5 mM catalyst) and EP (10 mM, 5 mM catalyst), respectively, at $T = 90$ °C and $p = 30$ bar $pH_2$ pressure after 7 s total hydrogenation time (i.e., 5 s bubbling plus a 2 s delay). For these results, deuterium-labeled unsaturated metabolite esters, which became available only recently[43,61,62] were essential, as in the presence of non-$pH_2$-nascent $^1$H the SOT efficiency is reduced[67]. Additionally, these results were preceded by extensive optimization and analysis as described in the following.

### Maximizing $^{13}$C hyperpolarization

The polarization process within PHIP-SAH involves two main stages: hydrogenation and SOT. In our investigation using VA, we examined the impact of various hydrogenation reaction parameters on $^{13}$C polarization, including reaction temperature $T$, $pH_2$ pressure $p$, and hydrogenation time $t_h$ (Fig. 4b–d) as well as catalyst concentration $c_{cat}$ (Supplementary Fig. 4), and unsaturated precursor concentration $c_{VA}$ (Supplementary Fig. 6). Unless otherwise specified, the standard parameters are $T = 90$°C, $p = 25$ bar, $c_{VA} = 80$ mM, $c_{cat} = 5$ mM and $t_h = 7$ s.

When we varied the temperature from 30 to 90 °C, we observed a progressive increase in $^{13}$C polarization, reaching its peak at temperatures above 70°C (Fig. 4b), with polarization values at 70°C and higher being within the standard deviation. Similarly, as we adjusted the $pH_2$ injection pressure across a range of 10 to 32 bar, we observed an increase in polarization, with the maximum achieved above 25 bar, where values were again equal within the standard deviation (Fig. 4c). Note that the solubility of hydrogen in solutions demonstrates a linear relationship with pressure and is approximately 4.5 mM/bar at 50°C for acetone[74]. Hence, it was concluded that elevated reaction temperatures and pressures play pivotal roles in achieving high polarization of contrast agents using our setup.

We varied $t_h$ from 5 to 70 s and discovered that the maximum $^{13}$C polarization of 80 mM EA occurred around 7 s (Fig. 4d). Beyond this time, further hydrogenation of the residual unsaturated precursor was dominated by relaxation of the $pH_2$-derived $^1$H spin order. We examined the data closer by fitting Eq. 2 and obtained the following fit parameters along with their associated errors: $P_{max} = (14.73 \pm 0.17)$%, $t_0 = (1.99 \pm 0.03)$ s, $T_{hydr} = (0.44 \pm 0.01)$ s, and $T_{relax} = (68.4 \pm 2.7)$ s, with $T_{relax}$ the $^1$H relaxation of $pH_2$-derived double spin order before SOT.

As we increased the concentration of VA from 10 to 160 mM while maintaining an unsaturated precursor-to-catalyst ratio of approximately 10, we observed a gradual decrease of the polarization (Supplementary Fig. 6) which was estimated under the assumption of complete conversion of VA to EA, which was likely not the case, especially at higher concentrations. This incomplete chemical conversion resulted in seemingly lower $^{13}$C polarizations for higher concentrations, where actually the chemical conversion was not fully realized; probably because of lack of diluted $H_2$. The influence of the catalyst concentration on $^{13}$C polarization was investigated to determine its role in the hyperpolarization process. Notably, within the range of 1 to 20 mM, the concentration of the catalyst demonstrated minimal impact on

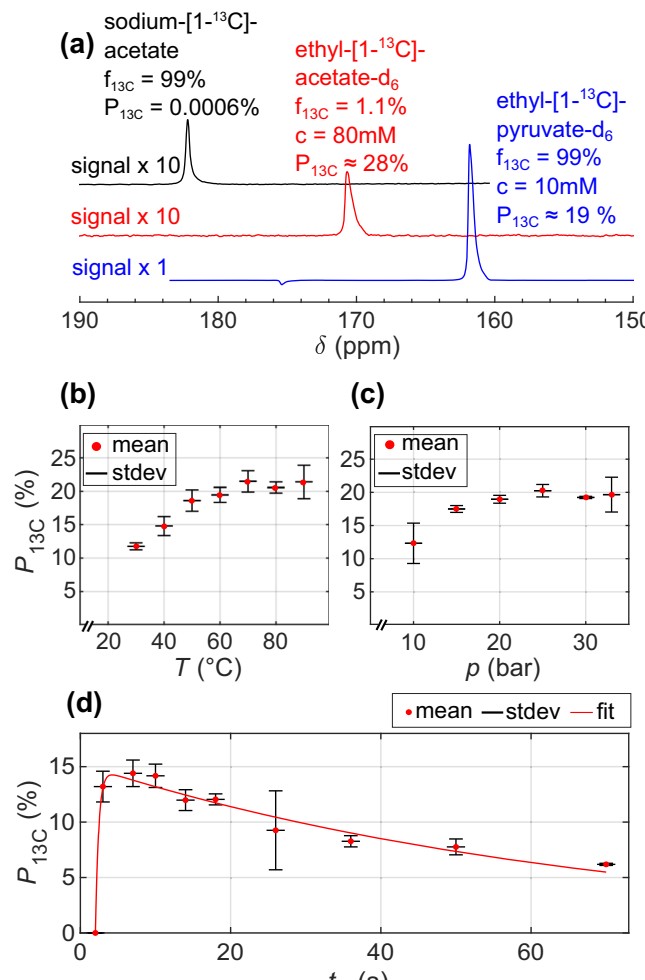

**Fig. 4 | Optimization to achieve high $^{13}$C hyperpolarized solution. a** $^{13}$C NMR spectra of a thermally-polarized reference solution (black) and hyperpolarized EA (red) and EP (blue). $^{13}$C-HP of EA and EP was quantified to 28% and 19%, respectively. The $^{13}$C polarization $P_{13C}$ for EA was optimized as function of temperature $T$ (**b**), $pH_2$ pressure $p$ (**c**), and hydrogenation time $t_h$ (**d**) using VA (mean, red dots, and stdev, black error bars, of $N = 3$ experiments each). The reaction kinetics were fitted with a described model (eq.-2). Unless stated otherwise parameters were set constant at $T = 90$ °C, $p = 25$ bar, $c_{VA} = 80$ mM, and $t_h = 7$ s. The different polarization levels observed in (**b**)–(**d**) are due to the typical inter-day reproducibility in SAMBADENA experiments (a factor of $1 \pm 0.2$[55,59], which is similar to state-of-the-art dDNP)[81]. Additionally, in (**d**) $pH_2$ with a lower enrichment was used.

the achieved $^{13}$C polarization indicating that the catalyst is not a limiting factor. This was particularly evident as all values at or exceeding a catalyst concentration of 2 mM were consistent within error intervals (Supplementary Fig. 4). Hence, for highly concentrated unsaturated precursors, optimal $t_h$ represents a tradeoff between hydrogenation time and relaxation time; in agreement with previous reports[55,68,75].

The ESOTHERIC sequence features a remarkable theoretical transfer efficiency, approaching nearly 100%, for converting the $^1$H two-spin order from the ethyl ester to the 1-$^{13}$C metabolite site[76]. In theory, this implies the potential for nearly 100% $^{13}$C polarization, provided that pure $pH_2$ is utilized and relaxation effects are negligible. However, practical considerations must also account for imperfections in the MRI system, including static field inhomogeneity and errors in pulse flip angles. A strategy to mitigate losses resulting from these MRI system imperfections is incorporating multiple 180° composite pulses[77]. In our investigations here, we determined that the maximum $^{13}$C polarizations were achieved when two 180° composite pulses

were applied during each free evolution interval of the SOT sequence (as depicted in Fig. 4b and detailed in Supplementary Fig. 5). As found in two subsequent HP experiments evaluating [1]H and [13]C polarization, this sequence allowed for the conversion of the initial [1]H polarization $P_{1H} \approx 30\%$ achieved after 7 s of hydrogenation of an 80 mM VA into $P_{13C} \approx 20\%$ [13]C polarization (Supplementary Fig. 8) that corresponds to about 66% transfer efficiency.

After the analysis of hydrogenation parameters, it's worth noting that the highest [13]C polarizations achieved for EP were lower, at 19%, compared to those for EA at 28%, although EP had an unsaturated precursor concentration eight times lower than EA. Several factors may contribute to this observation. Firstly, the hydrogenation process was optimized for EA, which is commercially available, but we expect a similar trend of improved polarization with higher temperatures and pH2 pressure for EP. Secondly, prior studies have indicated that the hydrogenation rate for VP may be lower than that of VA under the same reaction conditions, potentially leading to less complete reaction within executed 7 s[78]. Lastly, we measured the [13]C longitudinal relaxation of EP and found a time constant $T_1$ of $22.4 \pm 5.28$ s (Supplementary Fig. 7), which is shorter than the 67 s reported under similar conditions in a recent paper[43,71]. Additionally, we observed side-arm cleavage forming [1-[13]C]pyruvate-$d_3$ over time. Both findings may be attributable to impurities in the samples stemming from the preparation of the reaction solution. In a recent study using VP from the same synthesis, this phenomenon was led back to the presence of water in the reaction solution[76]. Consequently, we attribute the lower [13]C polarizations primarily to a fraction of EP that had cleaved before the SOT had successfully transferred the pH2 spin order to the pyruvate [1-[13]C]site.

In summary, achieving maximum [13]C hyperpolarization relies on a short hydrogenation time, minimized relaxation effects, and a high transfer efficiency. Enhancing the hydrogenation process for faster turnover might involve optimizing pH2 dissolution, potentially by maximizing gas-liquid interfaces. Moreover, the pH2 used had an enrichment fraction of ≈90% right after the ortho-to-para conversion (Supplementary Fig. 3). If 100% pH2 were employed, the [13]C polarization levels for EA and EP would increase to approximately 31% and 21%, respectively, i.e. by a factor of 1.11. Furthermore, a comprehensive examination of the polarization transfer, especially under clinical field strengths within the SAMBADENA setup, warrants further investigation, and we intend to present this analysis in future work.

### [13]C imaging

Motivated by the substantial [13]C polarization levels achieved at concentrations suitable for in vivo imaging, we embarked on demonstrating the rapid [13]C imaging capabilities. Utilizing HP EP prepared less than eight seconds

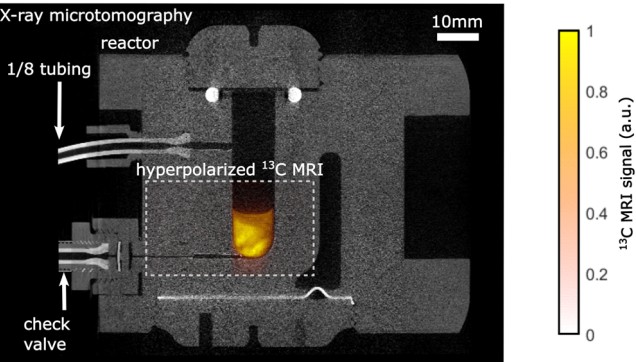

**Fig. 5 | Overlayed MRI and μCT image.** HP [13]C MRI of 10 mM ethyl [1-[13]C] pyruvate-$d_6$ (color) superimposed on an X-ray micro-tomography (μCT, gray) of the reactor. For the [13]C MRI, the signal was plotted on a red-to-yellow color scheme (Matlab: "autumn") and red was set transparent. Note that the two images were acquired at different times in different imaging devices and that the reactor and water-heating chamber were drained for μCT.

prior, we conducted a sub-second [13]C MRI experiment. The results revealed an intense [13]C signal (Fig. 5). Coregistration with a μCT image of the reactor provided spatial context, clearly indicating the signal's origin from the lower region of the reaction chamber and effectively picturing its contours.

### Outlook: towards obtaining HP metabolites in biocompatible aqueous solutions

In this study, we have not yet addressed the side-arm cleavage and purification steps of the PHIP-SAH procedure required to obtain HP acetate and HP pyruvate in aqueous formulations suitable for preclinical applications. The current acetone solutions with rhodium-based catalyst heated to >60°C are unsuitable for in vivo applications. While elevated temperatures facilitate fast hydrogenation and potentially fast side-arm cleavage reactions, the final agent solution needs to be adapted to a physiological temperature range for safe administration.

We have previously demonstrated the rapid and reliable adaption of temperature during the administration of HP aqueous solution of xenobiotic HEP to $(35 \pm 1)$ °C via heat exchange with injection syringe and catheter[56]. However, the PHIP-SAH experiment is more complex than our previous PHIP experiments, hence posing a significant yet unresolved obstacle.

Methods for rapid side-arm cleavage and purification are available in the literature and they may be adapted for our approach[31,53]. Techniques such as rapid base addition, solvent evaporation of acetone, and catalyst removal through filtration[30,53] seem promising for our approach. Additionally, phase separation using non-polar solvents like dichloromethane or chloroform may enable the separation of an aqueous phase containing the hydrolyzed contrast agents[40,48,79]—note that our presented setup is compatible with these solvents as well. In the future, a combination of these purification methods, potentially along with additional metal scavenging[80], may enable the production of highly pure and safe injection solutions.

During the work-up of the solution, the precious signal enhancement inevitably decreases due to $T_1$ relaxation. For instance, recent publications reported a factor of 0.5 of the starting [13]C hyperpolarization remaining after the work-up procedures[32,43]. Considering these values, our current levels would enable >10% [13]C polarization in the final HP agent solutions, which is sufficient for in vivo [13]C metabolic imaging[40]. Nonetheless, improving the [13]C polarization levels further is desirable and motivates more thorough future investigations of polarization transfer under MRI setup conditions.

For the SAMBADENA technique, side-arm cleavage, purification, and adaption of the agent solution will need to be incorporated within the magnetic field of the MRI system, while leaving space for the rodent, which particularly is not a trivial task. Purification at high magnetic fields, either at the fringe or inside the magnet, may benefit from extended $T_1$ relaxation times[30]. However, in the bore of a preclinical small rodent MRI setup as utilized in this study, this appears an engineering challenge. As part of our ongoing efforts, we are also exploring the implementation of SAMBADENA on a clinical MRI system. While this setting provides more room for sample hyperpolarization and purification, it introduces new complexities related to scaling the setup, positioning the reactor and accommodating the purification apparatus, and adapting to varying MRI specifications.

### Conclusion

SAMBADENA offers a compact footprint, fast hyperpolarization and administration and hence, holds promise for future applications. Although the implementation of a complete PHIP-SAH procedure including side-arm cleavage and purification presents challenges on both preclinical and clinical MRI systems, they appear to be solvable obstacles that merit exploration, especially now that we have demonstrated the efficient in situ hyperpolarization of pyruvate and acetate precursors within an MRI environment. Considering the low (additional) cost, small footprint[55], high sample throughput[59], and fast in vivo administration[56], SAMBADENA holds great promise to accelerate the translation of PHIP to lower the translational burden and enable widespread use of HP for metabolic MRI.

## Data availability

The data that support the findings of this study are available from the corresponding author upon reasonable request.

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

## Acknowledgements

This work was supported by the German Cancer Consortium (DKTK), the DKTK Joint-Funding Project "HYPERBOLIC", B.E.S.T. Fluidsysteme GmbH I Swagelok Stuttgart, the Preclinical Imaging Research Center - Core Facility (AMIR$^{CF}$; DFG-RIsources N° RI 00052) with lab and imaging systems (INST 39/1225-1) access, the Research Commission of the University Medical Center Freiburg, the German Research Foundation (SCHM 3694/1-1, SCHM 3694/2-1, SFB1479, HO-4604/2, HO-4604/3, PR 1868/3-1, PR 1868/5-1), and the German Federal Ministry of Education and Research (BMBF) in the funding program "Quantum Technologies-from Basic Research to Market" under the project "QuE-MRT" (contract number: 13^^N16448) and Juniorverbund 01ZX1915C (A.N.P). A.B.S and E.Y.C thank Wayne State University for Postdoctoral Fellow award. E.Y.C thanks the support of NSF CHE-1905341, and NIBIB R21 EB033872.

## Author contributions

Investigation: O.M., H.d.M., M.H. Data curation: O.M. Funding acquisition: A.N.P., L.S., E.Y.C., R.H., J.-B.H., D.v.E., A.B.S. Conceptualization: J.-B.H., A.B.S. Methodology: O.M., H.d.M., A.B.S. Resources: A.N.P., A.B., R.H., J.-B.H., M.Z., D.v.E., A.B.S. Supervision: M.Z., D.v.E., A.B.S. Visualization: O.M., H.d.M., A.B.S. Discussion of results: all authors. Writing original draft: O.M. and A.B.S. Review & editing: all authors.

## Funding

## Competing interests

E.Y.C. declares a stake of ownership in XeUS Technologies, LTD. E.Y.C. holds stock of Vizma Life Sciences, and serves on the scientific advisory board (SAB) of VLS. R.H., A.B., A.N.P., and J.-B.H. have filed an international patent application (patent name: Process for the production of ketocarboxylic acid vinyl esters, number: WO2023062106, author: R.H., R.B., A.N.P., and J.-B.H., geographical region: worldwide). R.H. is one of the founders and shareholder of QuantView GmbH, which commercializes the [1-$^{13}$C] vinyl pyruvate-$d_6$ and other vinyl esters of alpha-ketocarboxylic esters. All other authors declare no competing interest for both financial and non-financial.
