## [Peer Review file · Communications Chemistry]

This manuscript has been previously reviewed at another journal. This document only contains reviewer comments, rebuttal and decision letters for versions considered at Communications Chemistry.

Rapid In Situ Carbon-13 Hyperpolarization and Imaging of Acetate and Pyruvate Esters without External Polarizer

Corresponding Author: Dr Andreas Schmidt

Version 0:

Reviewer comments:

Reviewer #1

(Remarks to the Author)

The authors have made great efforts to improve the manuscript and have complied with the requests of the reviewers to an extent that this manuscript is now suitable for publication.

Reviewer #2

(Remarks to the Author)

see attached document

Reviewer #3

(Remarks to the Author)

The authors have extensively revised the manuscript, to clarify what was achieved exactly, and to describe the methods and results accurately. Their work is a notable step along a possible route to parahydrogen based hyperpolarised metabolic imaging, and will be of interest to the readership of Communications Chemistry.

Manuscript number: COMMSCHEM-24-0293-T

MS Type: Article

Title: "Rapid In Situ Carbon-13 Hyperpolarization and Imaging of Acetate and Pyruvate Esters without External Polarizer"

Correspondence Author: Dr. Andreas Benjamin Schmidt

Authors' response(General) : We thank the reviewers for the evaluation of our revised manuscript. We are glad that Reviewer 1 and 3 suggest publication in the current state and find our work of interest to the readership of Communications Chemistry.

We appreciate the positive evaluation by Reviewer 2 and value their additional feedback to improve our manuscript further. We agree with all new issues raised and have addressed them in our newly revised manuscript. Please find below a summary of the changes made in response to the reviewers' comments.

Reviewer R1 (General comment):

R1: The work authored by Mohiuddin et al. concerns the design, test and validation of a PHIP setup for hyperpolarized MRI small enough to fit inside the bore of a preclinical MRI scanner. This new design, as the title suggest, allows to "get rid of the polarizer" because, after hydrogenation, the polarization transfer sequence happens in the same magnetic field of scanner, thus reducing the footprint and complexity of the apparatus (no need of external magnet). The authors tested their method on two key molecules in the field of HP MR i.e. acetate and pyruvate in their ester form. The paper is clear and well-written. Nevertheless, some have to addressed prior o publication in this journal. Below you can find my comments divided according to the structure of the paper.

Authors' response: We thank the reviewer for their kind acknowledgement and taking the time to evaluate the manuscript the second time and providing a list of helpful changes required to improve the quality of the manuscript. Please find below our response to the queries raised by the reviewer.

R1.1: Abstract

Good and concise, but please specify the molecule concentration at which the 30 % ^{13}C pol was achieved, that is 10 mM.

A1.1: Authors' response: We thank the reviewer for the suggestion.

Changes made to the manuscript: As suggested by the reviewer, the concentrations of the molecules corresponding to the polarization levels have been added to the text:

"[...] Hyperpolarized ^{13}C MRI visualizes real-time metabolic processes *in vivo*. In this study, we achieved high ^{13}C polarization *in situ* in the bore of an MRI system for precursor molecules of most widely employed hyperpolarized agents: $[1-^{13}\text{C}]$ acetate and $[1-^{13}\text{C}]$ pyruvate ethyl esters in their perdeuterated forms, enhancing hyperpolarization lifetimes, hyperpolarized to $P_{13\text{C}} \approx 28\%$ at 80 mM concentration and $P_{13\text{C}} \approx 19\%$ at 10 mM concentration, respectively. [...]"

R1.2: Introduction - Page 1, second paragraph “to detect” instead of “to detecting”

A1.2: Authors’ response: We thank the reviewer for pointing out a mistake in the text. It has been corrected.

Changes made to the manuscript:

“[...]”

However, the intrinsically low sensitivity of magnetic resonance has been a persistent challenge, limiting MRI mostly to detect ^1H nuclei in water and lipids that are highly abundant in the mammalian body [...].”

R1.3: - Page 2, second paragraph remove “predominantly”. As of today, no clinical trials have been performed with techniques other than dDNP.

A1.3: Authors’ response: We acknowledge the comment made by the reviewer, and, as suggested, have removed the word “predominantly” from the sentence.

Changes made to the manuscript:

“[...]”

Biomedical applications, including all clinical studies, have ~~predominantly~~ relied on dissolution Dynamic Nuclear Polarization (d-DNP), which is recognized as the most established method for producing a variety of ^{13}C HP metabolites. [...].”

R1.4: - Page 2, second paragraph. I can understand that 1h from sample preparation to injectable solution could be considered time consuming (although in the clinic takes more than 1 h to prepare a patient for an HP MR experiment). Differently, I don’t see how preparing a sample for DNP is more elaborated than preparing one for PHIP. The first has the cryogenic to deal with, the second the production and supply of para-hydrogen and non trivial chemistry to synthesize the molecule’s precursor. Please consider rephrasing.

A1.4: Authors’ response: We appreciate the reviewer’s insightful comment. We agree that, while the equipment and sample preparation requirements for the two techniques are fundamentally different, both DNP and PHIP encounter unique and shared challenges. For PHIP, the primary difficulties lie in chemistry-related aspects, such as reactor design, reaction efficiency, synthesis of isotope-labeled precursors, and fast chemical work-up for purification. As highlighted in the manuscript, the “faster, more flexible, and more cost-effective production of HP media” is indeed a key research focus for both dDNP and PHIP communities.

Changes made to the manuscript: In response to the reviewer’s suggestion, we have removed the statement that dDNP is elaborate and avoided making judgments regarding sample preparation time. We have also restructured the final sentence to clarify that the challenges mentioned are being addressed in ongoing dDNP research as well.

“[...] The d-DNP hyperpolarization requires strong magnetic fields (>1T, typically employing superconducting magnets), cryogenic temperatures (<2K), and microwave irradiation to polarize agents in the solid phase. This is followed by rapid thawing and sample dissolution, typically using superheated water, to transfer HP metabolites into the liquid aqueous phase. This process often takes about one hour e.g. for producing a bolus of HP [1-¹³C]pyruvate. The impressive results achieved with HP MRI have sparked a persistent effort within the scientific community to explore other technologies and to enhance d-DNP for a faster, more flexible, and more cost-effective production of HP media.^{21–26}

[...]”

R1.5: -Page 3, third paragraph. Please consider rephrasing the argument about cost effectiveness as well. Most of the time, the PHIP community compares a PHIP home built setup to a SPINlab (2M \$, clinically ready machine). Let's be fair and take the 2 companies that can provide preclinically ready reliable machines: Polarize for dDNP and NVISION for PHIP. Polarize quotes its dDNP polarizer 650k € with a price per dose (30 mg of 1-¹³C-pyruvate + trityl) that is approx. 15 €. NVISION quotes its PHIP polarizer 400 k € with a price per dose that is 150 €. If we run 10 animals a day for 200 days per year, the running costs of dDNP is 30 k €, the one for PHIP is 300 k €. All that to say that real cost effectiveness must be taken as grain of salts. This is a scientific paper, let's talk science.

A1.5: Authors' response: We thank the reviewer for this valuable comment. We agree that comparing the cost of the two methods is challenging. For instance, while our home-built PHIP setup costs approximately €1,000 for materials, the main expenses for PHIP arise from the aforementioned chemistry challenges. Reducing synthesis costs will indeed be a crucial aspect for future commercialization.

Changes made to the manuscript: In light of the reviewer's feedback, we have removed the statement that PHIP is less expensive than dDNP from the paragraph and instead emphasized the features of PHIP:

“[...] Parahydrogen-based techniques hyperpolarize in the liquid state using the readily available spin order of the singlet isomer of molecular hydrogen (parahydrogen, pH₂), often employing inexpensive hardware and non-elaborate sample preparation methods.^{24,25} [...]”

R1.6: Material and Methods

good

A1.6: Authors' response: We kindly thank the reviewer for acknowledging the quality of our work in this section.

Changes made to the manuscript: no change requested by the reviewer.

R1.7: Results and Discussion

- The first sentence of the results is not supported by data. From Sup Figure 6 we see that 28% ¹³C pol was achieved in on EA for a concentration of 10 mM and not 80 mM. Please correct the polarization or concentration value.

A1.7: Authors' response: We appreciate your query and would like to clarify the confusion between Supplementary Figure 6 and main-text Figure 4. While both figures show ≈30% polarization for EA,

they correspond to different experiments. Supplementary Figure 6 presents data for 10 mM EA, whereas Figure 4 in the main text shows results for 80 mM EA. As stated in the first sentence of the Results and Discussion section, the EA concentration here is indeed 80 mM.

Changes made to the manuscript: no change requested by the reviewer.

R1.8: - The sentence about the efficiency of the ESOTHERIC sequence and the theoretical 100% ^{13}C is not supported neither by data nor references. Unless the authors provide one of the two, I would suggest removing it.

A1.8: Authors' response: We thank you pointing out a missing reference in the sentence. The reference [[10.1038/s41598-022-22347-1](https://doi.org/10.1038/s41598-022-22347-1)] has been added to the text.

Changes made to the manuscript:

“[...]”

The ESOTHERIC sequence features a remarkable theoretical transfer efficiency, approaching nearly 100%, for converting the ^1H two-spin order from the ethyl ester to the $1\text{-}^{13}\text{C}$ metabolite site.⁷⁹

[...]”

R1.9: -In the ^{13}C imaging part I would describe the signal as “intense” rather than “robust”

A1.9: Authors' response: We thank the reviewer for the recommendation and we agree on the remark

Changes made to the manuscript:

“[...] Motivated by the substantial ^{13}C polarization levels achieved at concentrations suitable for *in vivo* imaging, we embarked on demonstrating the rapid ^{13}C imaging capabilities. Utilizing HP EP prepared less than eight seconds prior, we conducted a sub-second ^{13}C MRI experiment. The results revealed an intense ^{13}C signal (Figure 5). [...]”

R1.10: - The sentence about the 100% pH₂ that would increase the polarization is not a result, it is a claim. Nothing against it, but it should be moved to Conclusion and Perspectives.

A1.10: Authors' response: We thank the reviewer for this comment. We agree that the discussion about polarization levels with pure parahydrogen is a projection rather than a direct result. However, we believe it is more appropriately discussed in the Discussion section rather than the Conclusion section. Nevertheless, we are open to moving it to the Conclusion if the editors of Communications Chemistry prefer this adjustment.

Changes made to the manuscript: no changes were made.

R1.11: - To me, the most interesting result is the dependence of the ^{13}C polarization on the reaction pressure, temperature and hydrogenation time. While the authors provide a sound explanation about the dependence on *th* very little is said concerning T and P. Looking at Fig 4b and 4c, it looks like there are 2 optima, i.e. 70°C and 25 bar. Could the authors comment on the presence of this maximum in the 2 curves?

A1.11: Authors' response: We thank the reviewer for this observation. We, too, noted the apparent maxima at 70°C and 25 bar. However, the polarization values obtained at 70°C and 25 bar and higher were equal within the standard deviation, meaning that the differences observed are not statistically significant. For this reason, we decided not to discuss these findings in detail, as the variations do not carry sufficient statistical relevance. We believe focusing on the broader trends is more appropriate, but we remain open to further exploration in future work.

Additionally, we suspect that at higher pressures, hydrogenation may have completed more quickly, allowing more time for relaxation in our experiments, which could explain the observation of seemingly lower polarization levels at those conditions. I.e., shorter hydrogenation times may be employed along with higher pressures in the future to improve P_{13C} .

Changes made to the manuscript: We clarified the explanation in the manuscript as follows:

“[...] When we varied the temperature from 30 to 90 °C, we observed a progressive increase in ^{13}C polarization, reaching its peak at temperatures above 70°C (**Figure 4b**), with polarization values at 70°C and higher being within the standard deviation. Similarly, as we adjusted the pH_2 injection pressure across a range of 10 to 32 bar, we observed an increase in polarization, with the maximum achieved at approximately 25 bar and beyond, where values were again equal within the standard deviation (**Figure 4c**). [...]”

R1.12 -Multiple refocusing pulses are suggested as a strategy to mitigate field inhomogeneity. Nevertheless, this study was performed with the reactor in the isocenter of the magnet. What happens, sequence wise, when the reactor will be in its real position (i.e. attached to the bed, but with the bed in the isocenter)?

A1.12: Authors' response: We thank the reviewer for an interesting query. The position of the reactor depicted in Figure 1 of the main text represents its real position at the isocenter. Even in animal studies, the reactor remains at the isocenter throughout the SOT sequence and only moves out once the sequence is complete and animal imaging begins.

Previous changes made to the manuscript based on similar question:

We clarified these details in our introduction: “[...] In our previous work, we demonstrated the administration and *in vivo* imaging within seconds using $[1-^{13}C]$ hydroxyethyl propionate- d_3 (HEP), a xenobiotic molecule hyperpolarized in water and administered without further purification.⁵⁴ We showed that both the reactor and the mouse can be positioned at the isocenter of the MRI system for hyperpolarization and *in vivo* imaging, respectively, facilitated by a motorized slider moving the mouse bed and reactor in between the two experimental steps during the tail vein administration. [...]”

General comment

The work authored by Mohiuddin et al. concerns the design, test and validation of a PHIP setup for hyperpolarized MRI small enough to fit inside the bore of a preclinical MRI scanner. This new design, as the title suggest, allows to “get rid of the polarizer” because, after hydrogenation, the polarization transfer sequence happens in the same magnetic field of scanner, thus reducing the footprint and complexity of the apparatus (no need of external magnet). The authors tested their method on two key molecules in the field of HP MR i.e. acetate and pyruvate in their ester form.

The paper is clear and well-written. Nevertheless, some have to addressed prior o publication in this journal. Below you can find my comments divided according to the structure of the paper.

Abstract

Good and concise, but please specify the molecule concentration at which the 30 % ^{13}C pol was achieved, that is 10 mM.

Introduction

- Page 1, second paragraph “to detect” instead of “to detecting”

- Page 2, second paragraph remove “predominantly”. As of today, no clinical trials have been performed with techniques other than dDNP.

- Page 2, second paragraph. I can understand that 1h from sample preparation to injectable solution could be considered time consuming (although in the clinic takes more than 1 h to prepare a patient for an HP MR experiment). Differently, I don't see how preparing a sample for DNP is more elaborated than preparing one for PHIP. The first has the cryogenic to deal with, the second the production and supply of para-hydrogen and non trivial chemistry to synthesize the molecule's precursor. Please consider rephrasing.

-Page 3, third paragraph. Please consider rephrasing the argument about cost effectiveness as well. Most of the time, the PHIP community compares a PHIP home built setup to a SPINlab (2M \$, clinically ready machine). Let's be fair and take the 2 companies that can provide pre-clinically ready reliable machines: Polarize for dDNP and NVISION for PHIP. Polarize quotes its dDNP polarizer 650k € with a price per dose (30 mg of 1- ^{13}C -pyruvate + trityl) that is approx. 15 €. NVISION quotes its PHIP polarizer 400 k € with a price per dose that is 150 €. If we run 10 animals a day for 200 days per year, the running costs of dDNP is 30 k €, the one for PHIP is 300 k €. All that to say that real cost effectiveness must be taken as grain of salts. This is a scientific paper, let's talk science.

Material and Methods

good

Results and Discussion

- The first sentence of the results is not supported by data. From Sup Figure 6 we see that 28% ^{13}C pol was achieved in on EA for a concentration of 10 mM and not 80 mM. Please correct the polarization or concentration value.

- The sentence about the efficiency of the ESOTHERIC sequence and the theoretical 100% ^{13}C is not supported neither by data nor references. Unless the authors provide one of the two, I would suggest removing it.

-In the ^{13}C imaging part I would describe the signal as “intense” rather than “robust”

- The sentence about the 100% pH_2 that would increase the polarization is not a result, it is a claim. Nothing against it, but it should be moved to Conclusion and Perspectives.

- To me, the most interesting result is the dependence of the ^{13}C polarization on the reaction pressure, temperature and hydrogenation time. While the authors provide a sound explanation about the dependence on t_h , very little is said concerning T and P. Looking at Fig 4b and 4c, it looks like there are 2 optima, i.e. 70°C and 25 bar. Could the authors comment on the presence of this maximum in the 2 curves?

-Multiple refocusing pulses are suggested as a strategy to mitigate field inhomogeneity. Nevertheless, this study was performed with the reactor in the isocenter of the magnet. What happens, sequence wise, when the reactor will be in its real position (i.e. attached to the bed, but with the bed in the isocenter)?